# Molecular Subtypes Based on Genomic and Transcriptomic Features Correlate with the Responsiveness to Immune Checkpoint Inhibitors in Metastatic Clear Cell Renal Cell Carcinoma

**DOI:** 10.3390/cancers14102354

**Published:** 2022-05-10

**Authors:** ByulA Jee, Eunjeong Seo, Kyunghee Park, Yi Rang Kim, Sun-ju Byeon, Sang Min Lee, Jae Hoon Chung, Wan Song, Hyun Hwan Sung, Hwang Gyun Jeon, Byong Chang Jeong, Seong Il Seo, Seong Soo Jeon, Hyun Moo Lee, Se Hoon Park, Woong-Yang Park, Minyong Kang

**Affiliations:** 1Department of Urology, Samsung Medical Center, Sungkyunkwan University School of Medicine, Seoul 06531, Korea; astherjee@skku.edu (B.J.); ejseo09@skku.edu (E.S.); s2623.lee@samsung.com (S.M.L.); jaehoontasker.chung@samsung.com (J.H.C.); wan.song@samsung.com (W.S.); hyunhwan.sung@samsung.com (H.H.S.); hwanggyun.jeon@samsung.com (H.G.J.); bc2.jung@samsung.com (B.C.J.); seongil.seo@samsung.com (S.I.S.); seongsoo.jeon@samsung.com (S.S.J.); hyunmoo.lee@samsung.com (H.M.L.); 2Samsung Genome Institute, Samsung Medical Center, Seoul 06531, Korea; kyunghee.park@samsung.com (K.P.); woongyang.park@samsung.com (W.-Y.P.); 3Oncocross Ltd., Seoul 04168, Korea; 99yirang@oncocross.com; 4Department of Pathology, Hallym University Dongtan Sacred Heart Hospital, Hwaseong 18450, Korea; byeon.sunju@welovedoctor.com; 5Division of Hematology-Oncology, Department of Internal Medicine, Samsung Medical Center, Sungkyunkwan University School of Medicine, Seoul 06531, Korea; sh1767.park@samsung.com; 6Department of Health Sciences and Technology, The Samsung Advanced Institute for Health Sciences & Technology (SAIHST), Sungkyunkwan University, Seoul 06355, Korea

**Keywords:** renal cell carcinoma, immune checkpoint inhibitor, responsiveness, molecular features

## Abstract

**Simple Summary:**

Immune checkpoint inhibitors (ICIs), such as programmed cell death protein 1 (PD-1) blockade, have proven to be the most effective agents for the management of many cancer types. Although ICIs are the current standard of care for treating metastatic clear cell renal cell carcinoma (ccRCC), 40–60% of patients still have intrinsic resistance to ICIs across multiple clinical trials. Therefore, identifying optimal biomarkers that can predict either responders or non-responders to ICIs has been of tremendous importance. Here, we generated targeted sequencing and whole transcriptomic sequencing of 60 patients with metastatic ccRCC treated with ICIs. Moreover, transcriptomic analysis was integrated to identify molecular subtypes using a total of 177 tumor samples by merging our data and published data derived from the CheckMate 025 trial. Our results show that these molecular subtypes are associated with specific genomic alterations, distinct molecular pathways, and differential clinical outcomes in patients with metastatic ccRCC treated with ICIs.

**Abstract:**

Clear cell renal cell carcinoma (ccRCC) has been reported to be highly immune to and infiltrated by T cells and has angiogenesis features, but the effect of given features on clinical outcomes followed by immune checkpoint inhibitors (ICIs) in ccRCC has not been fully characterized. Currently, loss of function mutation in *PBRM1*, a PBAF-complex gene frequently mutated in ccRCC, is associated with clinical benefit from ICIs, and is considered as a predictive biomarker for response to anti-PD-1 therapy. However, functional mechanisms of *PBRM1* mutation regarding immunotherapy responsiveness are still poorly understood. Here, we performed targeted sequencing (*n* = 60) and whole transcriptomic sequencing (WTS) (*n* = 61) of patients with metastatic ccRCC treated by ICIs. By integrating WTS data from the CheckMate 025 trial, we obtained WTS data of 177 tumors and finally identified three molecular subtypes that are characterized by distinct molecular phenotypes and frequency of *PBRM1* mutations. Patient clustered subtypes 1 and 3 demonstrated worse responses and survival after ICIs treatment, with a low proportion of *PBRM1* mutation and angiogenesis-poor, but were immune-rich and cell-cycle enriched. Notably, patients clustered in the subtype 2 showed a better response and survival after ICIs treatment, with enrichment of *PBRM1* mutation and metabolic programs and a low exhausted immune phenotype. Further analysis of the subtype 2 population demonstrated that *GATM* (glycine amidinotransferase), as a novel gene associated with *PBRM1* mutation, plays a pivotal role in ccRCC by using a cell culture model, revealing tumor, suppressive-like features in reducing proliferation and migration. In summary, we identified that metastatic ccRCC treated by ICIs have distinct genomic and transcriptomic features that may account for their responsiveness to ICIs. We also revealed that the novel gene *GATM* can be a potential tumor suppressor and/or can be associated with therapeutic efficacy in metastatic ccRCC treated by ICIs.

## 1. Introduction

Immune checkpoint inhibitors (ICIs), such as programmed cell death protein 1 (PD-1) blockade, have proven to be the most effective agents for the management of many cancer types [1]. Despite the low tumor mutational burden of renal cell carcinoma (RCC), it has unique immunologic features, including high immune infiltration score and increased infiltration by cytotoxic CD8^+^ T cells, which are known to be associated with the response to PD-1 blockade [2,3]. In this context, recent phase III clinical trials, such as CheckMate 025, CheckMate 214, and KEYNOTE 426, showed that ICIs-based regimens significantly improved the objective response and survival outcomes compared to that of tyrosine kinase inhibitors, particularly in advanced clear cell RCC (ccRCC) [4,5,6,7]. Although ICIs are the current standard of care for treating metastatic ccRCC, 40–60% of patients still have intrinsic resistance to ICIs across multiple clinical trials. Therefore, identifying optimal biomarkers that can predict either responders or non-responders to ICIs have been of tremendous importance [8].

While PD-ligand 1 expression is a conventional biomarker for predicting responsiveness to ICIs across various types of malignancies, data on RCC have been heterogeneous, and the predictive value of PD-ligand 1 expression is not clinically practical yet [8]. In contrast to other solid tumors, tumor mutational burden and neoantigen load, which have been the commonly explored predictors for ICIs therapy, were not associated with clinical responses to PD-1 blockade in advanced ccRCC [4]. Additionally, no difference in survival outcomes according to the patterns of CD8^+^ T cell infiltration in ccRCC patients treated with anti-PD-1 was found [4]. Recently, several studies reported that a loss of function (LOF) mutation in a PBAF-complex gene *PBRM1*, that is commonly mutated in ccRCC, was associated with better clinical benefit (CB) from ICIs [4,9,10]. In this context, a comprehensive understanding of the molecular mechanisms of *PBRM1* mutation in patients with ccRCC treated with ICIs could be critical for the development of a novel biomarker and to help predict which patients are most likely to benefit from ICIs treatment.

Here, targeted sequencing and whole transcriptomic sequencing of 60 patients with metastatic ccRCC treated with ICIs were performed. Moreover, transcriptomic analysis was integrated to identify molecular subtypes using a total of 177 tumor samples and by merging our data and published data derived from the CheckMate 025 trial [4]. Our results show that these molecular subtypes are associated with specific genomic alterations, distinct molecular pathways, and differential clinical outcomes in patients with metastatic ccRCC treated with ICIs.

## 2. Materials and Methods

### 2.1. Patients

The data of 60 patients with metastatic ccRCC treated with ICIs, particularly as either first-line (*n* = 8) (combination of anti-CTLA4 (ipilimumab) or anti-PD-1 (nivolumab)) or second-line (monotherapy of nivolumab) (*n* = 52) therapy from 2017 to 2020 at Samsung Medical Center, were retrospectively collected. The Institutional Review Board of our center approved the use of human archival tissues for this study (IRB no. SMC 2020-03-063).

Therapeutic responses of ICIs were determined every 3 to 4 months of treatment using abdomen-pelvis and chest-computed tomography scans. The responses were classified as complete response (CR), partial response (PR), stable disease (SD), or progressive disease (PD), according to the RECIST 1.1 criteria [11]. The clinical benefit (CB), non-clinical benefit (NCB), or intermediate benefit (IB) classification was defined in a previous report [9]. Briefly, CB included patients with CR, PR, or SD with any reduction in tumors lasting at least 6 months. NCB was defined as patients who experienced PD and were discontinued from ICIs therapy within 3 months. All other patients were assigned to the IB group.

### 2.2. Targeted Sequencing Preprocess

Tumor tissues from 60 patients were used for targeted sequencing of 380 cancer-related genes (CancerSCAN Version 3.1, a targeted-sequencing platform designed at Samsung Medical Center) and extracted from formalin-fixed paraffin-embedded tissues. Most samples had a mean coverage of ~900× with coverage at hotspots well above the mean. The paired-end reads were aligned to the human reference genome (hg19) using BWA (Version 0.7.5). Then, SAMtools (Version 0.1.18), GATK (Version 3.1-1), and Picard (Version 1.93) were used for file handling, local realignment, and removal of duplicate reads, respectively. The base quality scores were recalibrated using the GATK BaseRecalibrator based on known single-nucleotide polymorphisms (SNPs) and indels from dbSNP138.

### 2.3. RNA-Sequencing Preprocess

To perform RNA sequencing using 60 metastatic ccRCC and 5 adjacent nontumor tissues, total RNA was extracted utilizing the RNA extraction kit (RNeasy Mini Kit, QIAGEN, Maryland, MD, USA), and RNA integrity was verified using a 2100 Bioanalyzer (Agilent, Palo Alto, CA, USA). The libraries for sequencing were generated using the QuantSeq 3′ Library Prep Kit (Lexogen Inc., Vienna, Austria) according to the manufacturer’s instructions and sequenced on a HiSeq 2000 system (Illumina, San Diego, CA, USA). The reads were mapped to the hg19 human reference genome using STAR with default parameters. The number of reads mapped to each gene was calculated using RNA-Seq by Expectation-Maximization. Data processing and analysis were performed using the R/Bioconductor libraries. To preprocess the transcriptome data, genes with zero values across samples were filtered out. The data were normalized by subtracting the average expression values of the adjacent nontumor tissues per gene and centering the expression values of each sample and gene. In addition, three ccRCC samples that were outliers with respect to the overall mean and standard deviation and two ccRCC samples without information on the *PBRM1* mutation were filtered.

### 2.4. Immunohistochemical Analysis

To perform immunohistochemistry using 51 of the 60 available tissues of metastatic ccRCC patients, the Tissue microarray (TMA) method was applied. Briefly, representative tumor tissues (2 mm diameter) were taken from individual paraffin-embedded tumors (donor blocks) and arranged in new recipient paraffin blocks (tissue microarray blocks) using a trephine apparatus. One core tissue was taken from each case. Sections with a thickness of 5 mm were cut from each TMA block, deparaffinized, and dehydrated for immunohistochemical (IHC) staining. Ventana XT Benchmark (Ventana Medical Systems, Oro Valley, AZ, USA) for GATM (1:100 dilution, Catalog #ab32936, Abcam PLC, Cambridge, England, UK) was used IHC staining. Membranous and cytoplasmic staining of GATM was evaluated, and immunoreactivity for GATM was scored as follows: diffuse (more and equal 50% of tumor cells showed immunoreactivity) and moderate to strong expression of GATM was regarded as high expression (2+), diffuse or focal (less than 50% tumor cells showed immunoreactivity) and weak expression of GATM was regarded as low expression (1+), and no expression of GATM was regarded as no (0). Specimens with either weak or high expression of GATM were classified as GATM-positive and specimens with no expression of GATM were considered as GATM-negative. IHC data were reviewed by a well-experienced pathologist (S.-j.B.) who was unaware of other clinical data.

### 2.5. GSEA (Gene Set Enrichment Analysis) and ssGSEA (Single-Sample GSEA)

To perform the GSEA, the Hallmark gene sets from the Molecular Signatures Database (MSigDB Version 7.0) were used [12]. ssGSEA was computed using the “GSVA” package [13].

### 2.6. Immune Cell Type and Immune Type

The proportion of immune cell types was calculated using CIBERSORTx [14]. The proportion of 10 immune cell types was calculated by aggregation (for example, the proportion of macrophages was aggregated by macrophages M0, M1, and M2). Immune subtypes, including active and exhausted, were conducted using the nearest template prediction algorithm based on the expression of active and normal stroma signatures [15].

### 2.7. The Signatures of Differentially Expressed Genes (DEGs)

To identify upregulated or downregulated genes using GSE102806, DEG sets from three conditions were calculated. DEG1 was calculated as shPBRM1 vs. shControl in replicate 1 of the 786-O cell line. DEG2 was calculated as shPBRM1 vs. shControl in replicate 2 of the 786-O cell line. DEG3 was calculated as a *PBRM1* mutation vs. *PBRM1* wild type in the A-704 cell line. DEG4 and DEG5 were calculated from merged data (*n* = 177). DEG4 was differentially expressed in subtype 2 (Appendix A). DEG5 was calculated as *PBRM1* mutation vs. *PBRM1* wild type. DEGs were subjected to *t*-test (GSE102806) and permuted *t*-test (merged data), and the cutoff options were *p* < 0.05, FDR < 0.05, and log2 fold differences > 0.5.

### 2.8. Statistical Analysis

DEGs were calculated using *t*-test and permutation *t*-test. The Kaplan–Meier method was used to estimate PFS and OS. Categorical variables between the two groups were compared using Fisher’s exact test. One-way ANOVA was performed for the three groups. Student’s *t*-test was performed for both groups. All statistical analyses were performed using the R software.

### 2.9. Validation Sets

To merge transcriptomic data for the classification of subtypes, CheckMate 025 data were obtained from Appendix A published in Nature Medicine by Braun et al. [4]. To validate our findings, datasets were obtained from the TCGA-KIRC and GEO websites (accessed date, 17 December 2020; accession numbers: GSE102806 and GSE105288).

### 2.10. Cell Culture and Treatments

786-O cells transfected with siScramble (siScr) or siPBRM1 (AM16708, Thermo Fisher, Waltham, MA, USA) were seeded before treatment at 60–80% confluent at the time of the experiment. Additionally, 0.5 µM actinomycin D (11805017, Thermo Fisher, Waltham, MA, USA) was used for the indicated time.

### 2.11. Real-Time qPCR

786-O or Caki cells were transfected with siScr or siPBRM1, and after 48 h, real-time RT-PCR assays were conducted following the manufacturer’s instructions. Furthermore, 0.5 μg of total RNA was used as templates for reverse transcription through the ReverTra Ace qPCR RT Master Mix (Toyobo Co., Ltd., Kodakara island, Japan) according to the manufacturer’s instructions. Real-time PCR analysis was performed using the QuantStudio system with SYBR Premix Ex Taq (Takara Co., Ltd., Otsu, Japan). GATM F-5′-CAC TAC ATC GGA TCT CGG CTT, GATM R-5′-CTA AGG GGT CCC ATT CGT TGT and USH1C F-5′-TTC CGG CAT AAG GTG GAT TTT C, USH1C R-5′-GTA CAT TCG CAG CAC ATC ATA GA.

### 2.12. Cell Migration Assays

A total of 786-O cells were transfected with siRNAs against the control, *PBRM1*, or *GATM*. After 24 h, the 786-O cells seeded on glass-bottomed dishes pre-coated with fibronectin (100 g/mL) were scratched. After the cells reached 90% confluence, the monolayer was scratched with a pipette tip and incubated with a medium containing 300 µM H_2_O_2_ for 12 h in a humidified CO_2_ incubator at 37 °C.

### 2.13. Colony Formation Assay

Cells were transfected with the indicated siRNAs and seeded at 1000–2000 cells/well in 6-well plates. After 24 h, the medium was replaced with 300 µM H_2_O_2_ or low-glucose medium (11966025 Thermo Fisher) and incubated. After 7 days, the cells were fixed and stained with crystal violet. Triplicate wells were used for each experiment.

## 3. Results

### 3.1. The Characteristics of Genomic Alterations in Patients with ccRCC Treated with ICIs

The baseline demographics of patients with metastatic ccRCC in our cohort are described in Appendix A. Additionally, treatment outcomes according to first- and second-line therapies are summarized in Appendix A. Next, to evaluate the genomic landscape of patients with metastatic ccRCC (*n* = 60) treated with ICIs, we focused on targeted sequencing data to identify recurrently mutated genes in our cohort and found 17 recurrently altered genes. The most commonly altered genes in this cohort were *VHL* (*n* = 34, 56.7%), *PBRM1* (*n* = 18, 30.0%), *SETD2* (*n* = 16, 26.7%), and *BAP1* (*n* = 12, 20.0%), which were generally similar to those previously reported for ccRCC (Figure 1A) [4,9]. Next, when gene-specific alterations were compared between the clinical benefit (CB) group and the non-clinical benefit (NCB) group, only the *PBRM1* mutation among the 17 recurrently altered genes was significantly enriched in the CB group (Fisher’s exact test, *p* = 0.03, odds ratio for CB = 3.67, 95% confidence interval (CI) = 0.98–14.69) (Figure 1B). As expected, patients with the *PBRM1* mutation had significantly prolonged overall survival (OS), not progression-free survival (PFS), compared to those with *PBRM1* wild type (log-rank test, *p* = 0.018) (Figure 1C). These results indicated that CB following ICIs therapy was more prominent in patients with metastatic ccRCC harboring the *PBRM1* mutation.

### 3.2. The Characteristics of Molecular Subtypes in Patients with ccRCC Treated with ICIs

To expand our understanding of the molecular phenotypes of metastatic ccRCC treated with ICIs, WTS data were generated using 177 tumor samples by merging our data and CheckMate 025 data. We aimed to identify molecular subtypes by utilizing four signatures associated with *PBRM1*-mutation and *PBRM1* LOF (Loss of Function). Two signatures of *PBRM1*-mutation were previously reported to be differentially expressed genes (DEGs) between patients with and without *PBRM1* mutations [16]. The signatures of *PBRM1* LOF also previously reported that high angiogenesis and less immunomodulation were related to phenotypes of *PBRM1* loss [17]. Unsupervised clustering analysis was performed based on the four signatures associated with *PBRM1* mutation and *PBRM1* LOF and identified three molecular subtypes in 177 patients (Figure 2, top). Patients with subtype 1 (*n* = 64, 36%) were characterized by the moderate expression of upregulated and downregulated genes related to the *PBRM1* mutation with relatively lower expression of angiogenesis and a mixed pattern of immunomodulatory signature. Additionally, 12 patients with subtype 1 showed *PBRM1* mutations (20%). Interestingly, patients with subtype 2 (*n* = 75, 42%) were characterized by consistent expression patterns of upregulated and downregulated genes related to the *PBRM1* mutation, relatively higher expression of angiogenesis, and relatively lower expression of immunomodulatory signature with higher mutation rate of *PBRM1* (*n* = 44, 72%), compared to those with subtypes 1 and 3. The low expressions of immunomodulatory were seen in subtype 2, consistent with previous finding that *PBRM1* loss are associated with a nonimmunogenic tumor phenotype [17]. Patients with subtype 3 (*n* = 38, 22%) were characterized by the inverse expression of both upregulated and downregulated genes related to the *PBRM1* mutation, moderate expression of angiogenesis, and enrichment of higher immunomodulatory signature. Only five patients had the *PBRM1* mutations (8%) in subtype 3 (Figure 2, top).

We further characterized the prevalence of the six commonly mutated genes by overlapping recurrently mutated genes from our data and CheckMate 025 data and found a higher prevalence of alterations such as *VHL* (*n* = 36, 48%), *PBRM1* (*n* = 44, 72%) and *SETD2* (*n* = 28, 57%) in patients in subtype 2 (Figure 2, middle) than that of the other two subtypes. Next, to evaluate the key biological features related to these molecular subtypes, pairwise comparisons of each subtype were performed. Unique upregulated DEGs were identified in each molecular subtype (permuted *t*-test; *p* < 0.05, false discovery rate (FDR) < 0.05, and log2 fold difference > 0.5; Appendix A). First, 397 upregulated genes unique to subtype 1 were enriched in biological process terms for cell–cell signaling (ES = 2.46, *p* = 4.39 × 10^−4^) and cell development (ES = 2.34, *p* = 2.15 × 10^−5^). Second, 1569 upregulated genes unique to subtype 2 with a high proportion of *PBRM1* mutations were associated with the metabolic process (ES = 25.93, *p* = 5.80 × 10^−42^) and xenobiotic metabolism (ES = 12.56, *p* = 7.04 × 10^−14^). Finally, 459 upregulated genes unique to subtype 3 showed the activation of genes related to the cell cycle (ES = 12.55, *p* = 3.62 × 10^−23^) and immune response (ES = 6.97, *p* = 1.19 × 10^−10^) (Figure 2, bottom). Taken together, the molecular stratification of 177 ccRCC tumors, treated with ICIs, was conducted into three subtypes with biologically distinct transcriptomes.

### 3.3. Subtype 2 Is Associated with Higher Metabolic Processes and Lower Exhausted Immune Types than the Other Two Subtypes

Next, when the treatment responses were examined according to each subtype, significant differences in the clinical responses and benefits were not found (Appendix A). To evaluate the prognostic relevance of each subtype, the PFS and OS were compared according to each subtype. Notably, subtype 2, compared to that of subtypes 1 and 3, was significantly associated with OS (*p* = 0.0042) (Figure 3A) but not with PFS (*p* = 0.381) (Appendix A). To further understand the molecular mechanisms of survival outcomes, transcriptomic pathway programs were explored by performing gene set enrichment analysis (GSEA) and single-sample GSEA (ssGSEA) using Hallmark gene sets (*n* = 50) in each subtype. Overall, 18 gene sets, including inflammatory response, oxidative phosphorylation, and E2F target pathways, were activated in each subtype (one-way analysis of variance (ANOVA) test, *p* < 0.0005, Figure 3B and Appendix A). Both subtypes 1 and 3 were activated by immune-related pathways (INFLAMMATORY_RESPONSE, COMPLEMENT, and IL6_JAK_STAT3). Subtype 3 differentiated from subtypes 1 and 2 through the enhanced activation of cell cycle progression pathways (G2M_CHECKPOINT, E2F_TARGETS, and MITOTIC_SPINDLE). Particularly, in patients with subtype 2, metabolic-related pathways (OXIDATIVE_PHOSPHORYLATION, FATTY_ACID_METABOLISM, and ADIPOGENESIS), the HYPOXIA and REACTIVE_OXYGEN_SPECIES pathways were activated. Previous reports demonstrated that ccRCC tumors with *PBRM1* mutations were activated with a hypoxic transcriptional signature, which is in agreement with our findings [9,17,18].

To further evaluate the immune-related characteristics of each subtype, immune cell fractions were analyzed using the CIBERSORTx deconvolution algorithm. When the proportion of diverse immune cell repertories was examined, no prominent differences in cell types were observed according to molecular subtype (Figure 3C, left). Interestingly, subtype 3 had a higher proportion of CD8 T cells, which are related to the mode of action of immunotherapy (Figure 3C, right). This result might be associated with the higher immunomodulatory activity in patients with subtype 3 (Figure 2).

Given the unexpected finding that subtype2 with high *PBRM1* mutation rate was not related to proportion of CD8 T cell, we investigated whether tumor intrinsic features affect the response to ICIs within the context of different immune subtypes. We therefore analyzed the immune subtypes classified as active immune and exhausted immune, which have been reported to play distinct roles in cancer progression and are associated with the patient’s clinical outcomes [15]. Prediction of the immune types in each tumor using the nearest template prediction algorithm revealed that subtype 3, with the worst survival, exhibited the highest proportion of exhausted immune subtype and the lowest proportion of active immune subtype compared to that of the other two molecular subtypes. However, subtypes 1 and 2 had similar proportions of the active immune subtype, whereas subtype 2 with good survival had half the proportion of exhausted immune subtype compared to that of subtype 1 (Figure 3D, left). Additionally, we assessed immune subtypes according to the *PBRM1* mutation and identified that patients with the *PBRM1* mutation showed a higher active immune subtype as well as a lower exhausted immune subtype compared to those without the mutation (Figure 3D, right). Moreover, the OS and PFS were assessed according to immune subtypes, including active immune, exhausted immune, and non-immune subtypes. We observed that the immune-activated subtype showed a tendency toward better PFS (*p* = 0.052, Figure 3E) and OS (*p* = 0.469, Appendix A), which was in agreement with previous results [15,19,20]. These results indicated that each molecular subtype had different immune subtypes, which may affect the tumor microenvironment related to the responsiveness of ICIs. Moreover, an exhausted immune transition from the active immune subtype in ccRCC may indicate the induction of an aggressive phenotype and an unfavorable prognosis after ICIs treatment.

### 3.4. GATM Expression Associated with PBRM1 Mutation as a Novel Biomarker of Therapeutic Response in Patients with ccRCC Treated by ICIs

Next, we aimed to identify key modulating genes associated with subtype 2 harboring a higher proportion of *PBRM1* mutations as well as showing the best survival outcomes. We analyzed cell line data (GSE102806), including 786-O and A-704 cell lines, which were associated with *PBRM1* LOF. First, we generated six gene sets, either upregulated or downregulated DEGs, by comparing treatment samples and control samples (see details in Section 2). To further evaluate the DEGs, including DEG1, DEG2, and DEG3, we applied the merged data of human samples (*n* = 177) using the DEGs and by performing ssGSEA and identified distinct expression patterns in each subtype (Figure 4A). Notably, subtype 2 showed a significantly higher expression of the upregulated DEGs and a lower expression of the downregulated DEGs compared with subtypes 1 and 3. Likewise, tumors harboring the *PBRM1* mutation exhibited a higher expression of the upregulated DEGs, whereas the downregulated DEGs were not related compared with that of the *PBRM1* wild type (Figure 4B and Appendix A). These results suggested that upregulated DEGs rather than downregulated DEGs derived from cell line data were clearly validated in data derived from human samples showing the distinct characteristics of *PBRM1* LOF.

Then, five DEGs were used (see details in Section 2), and 4 genes and 0 genes were identified as commonly upregulated or downregulated genes, respectively, by overlapping all five DEGs (Figure 4C, left and Appendix A). The four identified genes, *AMACR*, *SLC6A3*, *GATM*, and *USH1C*, were upregulated and potentially associated with *PBRM1* mutation (Figure 4C, right). Among them, we focused on *GATM* (*p* = 1.01 × 10^−5^) and *USH1C* (*p* = 1.94 × 10^−4^), which were significantly differentially expressed in tumors harboring the *PBRM1* mutation. In particular, the *GATM* gene showed a non-tumor-specific expression compared with that of the levels in tumor tissues in The Cancer Genome Atlas Kidney Renal Clear Cell Carcinoma (TCGA-KIRC) data (*n* = 607, *p* = 3.07 × 10^−7^; Appendix A) and GSE105288 (*n* = 43, *p* = 0.045; Appendix A). However, the *USH1C* gene showed tumor-specific expression compared with that of the levels of nontumor tissues in TCGA-KIRC data (*p* = 6.0 × 10^−9^; Appendix A) and GSE105288 (*p* = 0.503; Appendix A). We then evaluated the clinical relevance of these two genes. When compared according to the expression levels of each gene, there were no differences in DFS (disease-free survival) (Appendix A) or PFS (Appendix A). However, a higher expression of *GATM* was consistently correlated with clinical outcomes of OS in TCGA-KIRC data (*p* = 3.64 × 10^−7^; Figure 4D, left) and the merged data (*p* = 0.030; Figure 4D, right). A higher expression of *USH1C* was correlated with clinical outcomes of OS in TCGA-KIRC data (*p* = 0.00015; Figure 4D, left), whereas it did not reach statistical significance for OS in the merged data (*p* = 0.203; Figure 4D, right). Accordingly, GATM was prioritized as a potential driver in the following analysis.

### 3.5. GATM Protein Levels Using Immunohistochemistry Are Related to Favorable Survival

Next, to validate the prognostic role of GATM, we evaluated the GATM protein expression via IHC staining in 51 metastatic ccRCC patients and stratified it into two groups according to the status of GATM expression (GATM-positive and GATM-negative, respectively) (Figure 5A). Then, the association of GATM protein levels with survival outcomes was investigated. Notably, the GATM-positive group had the significantly better PFS (*p* = 0.0156; Figure 5B, top) and OS outcomes (*p* = 0.0013, Figure 5B, bottom) after ICIs therapy compared to GATM-negative group. We further analyzed GATM protein levels based on *PBRM1* mutation status and the molecular subtypes, and found that patients with *PBRM1* mutation exhibited a higher percentage of GATM-positive specimens than patients without the *PBRM1* mutation (Figure 5C, top). Additionally, we observed that subtypes 2 had a highest percentage of GATM-positive cases, whereas subtype 3, with aggressive phenotypes and poor survival, had the lowest percentage of GATM-positive specimens compared to that of subtype 1 and 2 (Figure 5C, bottom). These findings suggested that GATM protein levels are associated with *PBRM1* mutation status and less aggressive phenotypes and exhibit potential clinical utility as a prognostic marker for metastatic ccRCC with ICIs treatment.

### 3.6. PBRM1 Deficiency and GATM Upregulation in Stress Conditions Reduce Cell Proliferation

We examined whether *GATM* expression is regulated by the loss of *PBRM1* in ccRCC lines. *PBRM1* knockdown in Caki-1 and 786-O cells showed an increase in *GATM* or *USH1C* expression (Appendix A and Figure 6A). We also tested whether *GATM* expression was regulated by *PBRM1* at the transcriptional level using 0.5 µM actinomycin D (transcriptional inhibitor). As expected, increased levels of *GATM* transcripts were sustained upon *PBRM1* knockdown in 786-O cells treated with actinomycin D compared to that of control cells (Figure 6B). The results indicated that *GATM* expression was increased at the transcriptional level following *PBRM1* knockdown. *PBRM1* has been reported to protect cancer cells under high-stress conditions, and patients with ccRCC harboring *PBRM1* mutations show better responsiveness to PD-1 inhibitors [9,21]. Given the protective roles of *PBRM1* in stress response, we examined whether *GATM* induced by *PBRM1* deletion participated in the anticancer state under stress conditions. An in vitro wound healing assay was performed using 786-O cells. As shown in Figure 6C, 786-O cells transfected with siScr, siPBRM1, or siGATM were scratched and incubated with high concentrations of hydrogen peroxide (H_2_O_2_) for 12 h. *PBRM1* knockdown in the 786-O cells exposed to H_2_O_2_ led to little movement compared to that in control cells. Depletion of *GATM* in *PBRM1*-knockdown 786-O cells augmented the migration, and motility by *GATM* knockdown was similar to that in the control cells (Figure 6C). These data indicate that migration ability was lost by *PBRM1* loss-induced *GATM* upregulation.

Next, we examined the effect of *GATM* induced by *PBRM1* knockdown on the proliferation of 786-O cells under stress conditions such as that in H_2_O_2_ or low-glucose medium. *PBRM1*-depleted cells incubated with medium containing high concentrations of hydrogen peroxide or low glucose lost the ability to form colonies, which was recovered by silencing *GATM* (Figure 6D). The results suggested that the antiproliferative ability of the *PBRM1* mutation under stress conditions was accompanied by an increase in *GATM* expression.

More importantly, the association of *GATM* expression and *PBRM1* mutations with clinical prognosis was investigated; merged data (*n* = 177) derived from human samples were stratified into four groups according to the status of *PBRM1* mutation and *GATM* expression: (PBRM1_MUT+HIGH_GATM (*n* = 42), PBRM1_MUT+LOW_GATM (*n* = 19), PBRM1_WT+HIGH_GATM (*n* = 55) and PBRM1_WT+LOW_GATM (*n* = 61)). As expected, patients with the *PBRM1* mutation and high expression of *GATM* had the best PFS (*p* = 0.069; Appendix A) and OS outcomes (*p* = 0.0012, Figure 6E, left). The molecular subtypes of the four groups were further analyzed, and it was found that patients in the PBRM1_MUT+HIGH GATM group were enriched in subtype 2 (88%), and patients in the PBRM1_WT+LOW_GATM group were enriched in subtypes 1 (46%) and 3 (48%) (Appendix A). More importantly, multivariate analysis of the merged data also revealed the prognostic significance of *GATM* expression and *PBRM1* mutation (hazard ratio [HR] = 2.067, 95% confidence interval [CI], 1.147–3.726; *p* = 0.016; Table 1). On further analysis, *PBRM1* mutation and *GATM* expression were associated with significant OS (*p* = 9.40 × 10^−7^, Figure 6E, right) rather than PFS (*p* = 0.332, Appendix A) in the TCGA-KIRC cohort. Although the TCGA-KIRC cohort was predominately composed of non-ICIs-treated patients, the survival result suggested that patients with the *PBRM1* mutation and high *GATM* expression are consistently associated with superior OS. These retrospective analysis results collectively suggest that the status of the *PBRM1* mutation and *GATM* expression were considered as prognostic key factors of RCC patients.

The *GATM* gene encodes glycine amidinotransferase, catalyzing the rate-limiting step in the synthesis of creatine, which plays a pivotal role in cancer progression and immunotherapy. Previous studies have demonstrated that creatine inhibits the growth of tumor cells both in vitro and in vivo and is associated with the regulation of T cell antitumor immunity [22,23]. In this context, the creatine metabolism level was examined for each subtype (*p* = 2.63 × 10^−7^; Appendix A) as well as in patients with the *PBRM1* mutation vs. in patients without the *PBRM1* mutation (*p* = 0.048; Appendix A). Interestingly, patients with subtype 2 and *PBRM1* mutation showed significantly elevated creatine metabolism. Next, the correlation between creatine metabolism and *GATM* expression levels was tested, revealing that the *GATM* expression had a significantly positive correlation with creatine metabolism (correlation coefficient, r = 0.49, *p* = 4.61 × 10^−12^, Appendix A). Collectively, activation of *GATM* may represent an efficient therapeutic approach for ccRCC harboring the *PBRM1* mutation under ICIs by regulating creatine metabolism to enhance antitumor T cell immunity in the tumor microenvironment.

## 4. Discussion

This study reports a comprehensive molecular analysis of patients with metastatic ccRCC receiving ICIs to investigate the role of tumor genomic and transcriptomic features in determining the response and survival outcomes following ICIs treatment. Our findings are summarized in Figure 7.

The dynamics of genomic and transcriptomic features, including the *PBRM1* mutation, metabolic process, active or exhausted immune types, and *GATM* expression, contributed to the responsiveness and prognosis to immunotherapy in metastatic ccRCC. More importantly, for the first time, we found that *GATM* plays a suppressive role by linking the the *PBRM1* mutation to patients with ccRCC treated with ICIs. Our unsupervised transcriptomic analysis based on signatures related to the the *PBRM1* mutation and *PBRM1* LOF identified three molecular subtypes. This subtyping approach in this study showed concordance with previous reports on gene expression-based subgrouping in large RCC datasets [10,24,25,26]. Indeed, we found an association between molecular subtypes and differential biological profiles and different prognoses to ICIs in patients with metastatic ccRCC. Patients in subtype 2 demonstrated favorable OS with a higher proportion of them with the *PBRM1* mutation and who are enriched with angiogenesis and metabolic pathways. In contrast, patients in subtypes 1 and 3 showed worse clinical outcomes with a low proportion of them with the *PBRM1* mutation and who are downregulated with angiogenesis signature but upregulated with immune-related and cell-cycle pathways. Overall, the unique features of these subtypes provide their utility in understanding the differential prognosis and responsiveness to ICIs treatment.

One of our key findings is that subtype 2, with a higher proportion of the *PBRM1* mutation, showed enrichment of multiple metabolic pathways, including oxidation and phosphorylation, fatty acid metabolism, adipogenesis, and hypoxia pathways. In fact, hypoxia signaling as a master regulator is associated with the dysregulation of metabolic genes in RCC [27]. Indeed, previous studies have demonstrated that VHL loss and hypoxia-inducible factor stabilization are associated with the reprogramming of metabolic pathways in RCC [28,29,30]. In summary, our results not only validated previous findings on RCC metabolism but also further explored the metabolic differences among the three subtypes. Interestingly, in-depth transcriptome analysis revealed an unexpected inconsistency between the response to ICIs and immune cell types. Additionally, previous studies demonstrated that *PBRM1* loss shows a nonimmunogenic tumor phenotype associated with ICIs [17], and CD8 T cell infiltration of immunofluorescence in CheckMate025 was not associated with response to PD-1 blockade [4]. Therefore, we investigated whether the association between immune cell types and response to ICIs would be better characterized by considering immune types, including active and exhausted immune subtypes. We found that tumors harboring the the *PBRM1* mutation or molecular subtype 2, which had the best survival outcome, showed a lower percentage of exhausted immune subtype compared to that of tumors with *PBRM1* wild type or subtypes 1 and 3. Previous studies have also shown that high proportion of exhausted immune types in tumor samples results in poor prognosis and an aggressive phenotype [15,31,32]. Taken together, these data suggest that ccRCC tumors can undergo a transition in immune subtypes from active to exhausted types, which may affect the tumor microenvironment to be different response to ICIs.

Moreover, our data indicated that *GATM* is a potential gene-linking *PBRM1* mutation and *PBRM1* LOF. *GATM* encodes glycine amidinotransferase, a mitochondrial enzyme that catalyzes the transfer of guanidinoacetic acid, which is a substrate for creatine synthesis. Interestingly, several studies have demonstrated that creatine inhibits the growth of tumor cells both in vitro and in vivo [22,33,34]. Furthermore, in mouse cancer models, treatment with creatine, either through intraperitoneal injection or through oral administration, effectively suppressed tumor growth, which was associated with a significant reduction in the number of exhausted T cell phenotypes among the tumor-infiltrating CD8^+^ T cells [22]. The exact mechanism by which creatine or *GATM*-attenuated cancer growth and related to immunotherapy is still unclear; however, a possible tumor-suppressive role of *GATM* was supported by the fact that low expression of *GATM* was associated with poor survival of patients with ccRCC treated with ICIs (as shown in Figure 4D). In addition, the status of GATM protein expression was significantly associated with PFS and OS after ICIs treatment (as shown in Figure 5B). Therefore, the activation of *GATM* has the potential to become an effective approach for enhancing the efficacy of ICIs therapies.

We acknowledged the limitations of the present study. First, the sample size was relatively small, which is the most critical pitfall of our study. To overcome this drawback, data derived from the CheckMate 025 study were merged. Second, a retrospective study was performed and samples were collected; therefore, there was an unavoidable risk of bias, such as selection and misclassification biases. Third, most patients were treated with anti-PD1 monotherapy as a second-line treatment after failure of the first-line targeted therapy. Thus, there was a discrepancy in the time between tumor sampling and anti-PD1 therapy, which means that there was time gap between genomic and/or transcriptomic status and ICIs treatment response status Actually, the population of this study included either first-line therapy (*n* = 8) or more than second-line treatments that failed withTyrosine Kinase inhibitors (TKIs, *n* = 52). The selective pressure of multiple lines of TKIs treatment could increase the genomic complexity of tumors and tumor microenvironments, thus influencing responsiveness to ICIs therapy. Fourth, our data were based on WTS analysis, not single-cell sequencing, resulting in a loss of immune cell compartment, which is particularly important for understanding the immune microenvironment and limiting the ability to perform such an immune cell type proportion analysis. Finally, further experiments should be performed to establish causality between the *PBRM1* mutation and *GATM* expression and to determine how *PBRM1-GATM* is involved in the responsiveness to ICIs treatment.

## 5. Conclusions

This study provides critical insight into genomic and transcriptomic mechanisms such as *PBRM1* mutations, metabolic processes, immune subtypes (active or exhausted), and *GATM* expression that contribute to the response to ICIs therapy in patients with metastatic ccRCC. Moreover, our data underscore the prognostic importance of the *PBRM1* mutation and *GATM* expression in patients with metastatic ccRCC treated with ICIs. Moving forward, it will be important to validate these findings in future clinical trials and to investigate the mechanisms on the interaction between the *PBRM1* mutation and *GATM* expression in the context of ICIs responsiveness.

## Figures and Tables

**Figure 1 cancers-14-02354-f001:**
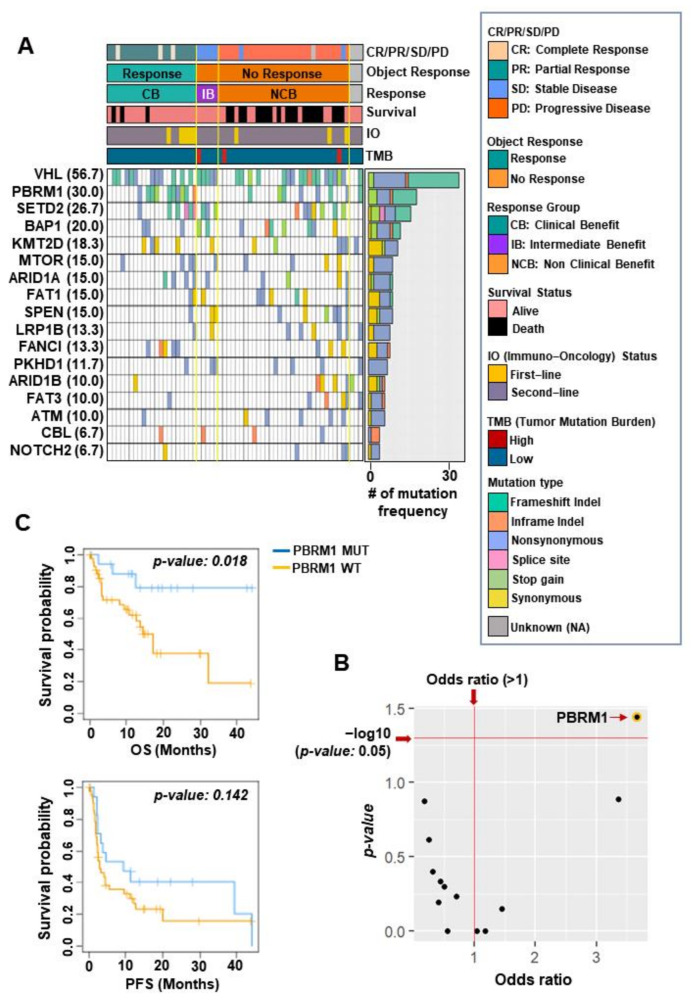
Overall mutational landscape in 60 patients with metastatic clear cell carcinoma treated with immune checkpoint inhibitors. (**A**) Heatmap shows 17 recurrently mutated genes in our cohort ordered by the number of mutation frequencies. (**B**) The plot shows the *p*-value and odds ratio for the 17 genes by performing Fisher’s exact test between the clinical benefit versus non-clinical benefit groups. (Red dashed lines denote *p* < 0.05 and odds ratio > 1. Each black dot denotes the 17 recurrently mutated genes). (**C**) Kaplan–Meier plots show the overall survival (top) and progression-free survival (bottom) of patients who did or did not harbor mutations in *PBRM1*. PBRM1 MUT, PBRM1 mutation (blue); PBRM1 WT, PBRM1 wild type (yellow).

**Figure 2 cancers-14-02354-f002:**
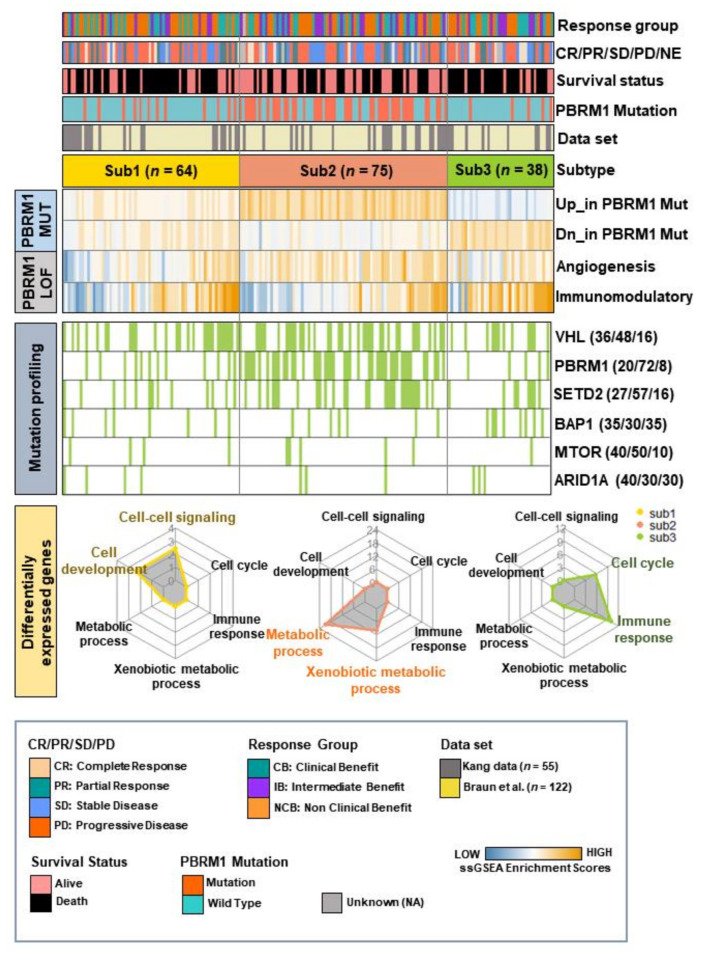
Characteristics of subtypes based on the gene sets associated with *PBRM1* mutation and *PBRM1* loss of function. Heatmap of the single-sample gene set enrichment analysis shows the enrichment scores of the four gene sets in the merged data set (top). Mutation plot shows six commonly mutated genes by overlapping recurrently mutated genes from both our data and the CheckMate 025 data (middle). (Green color denotes harboring mutation). Radar plots show significantly enriched gene ontology terms using upregulated genes in each subtype (bottom).

**Figure 3 cancers-14-02354-f003:**
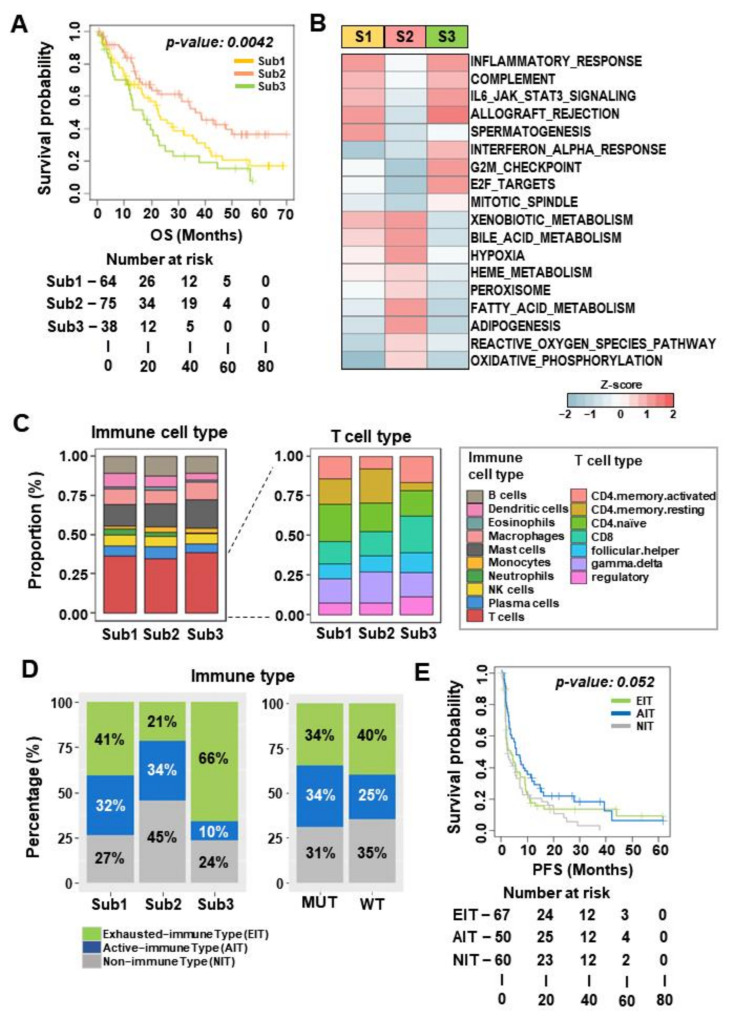
Patients in subtype 2 with enrichment of the *PBRM1* mutation are associated with metabolic pathways and transition of immune types. (**A**) Kaplan–Meier plot analysis of the overall survival for each subtype. (**B**) Heatmap of Hallmark gene sets. The mean Z score of the single-sample gene set enrichment analysis for each gene set was calculated (S1, Sub1; S2, Sub2; S3, Sub3). (**C**) CIBERSORTx findings show the proportion of distinct immune cell subpopulations (left) and the proportion of T cell subpopulations (right). (**D**) Barplots show the percentage of immune types, including active immune, exhausted immune and nonimmune types, in each subtype (left) and in the group with *PBRM1* mutation vs. the group without *PBRM1* mutation (right). (**E**) Kaplan–Meier plot analysis of progression-free survival for each immune type. AIT, active immune type (navy); EIT, exhausted immune type (green); NIT, non-immune type (gray).

**Figure 4 cancers-14-02354-f004:**
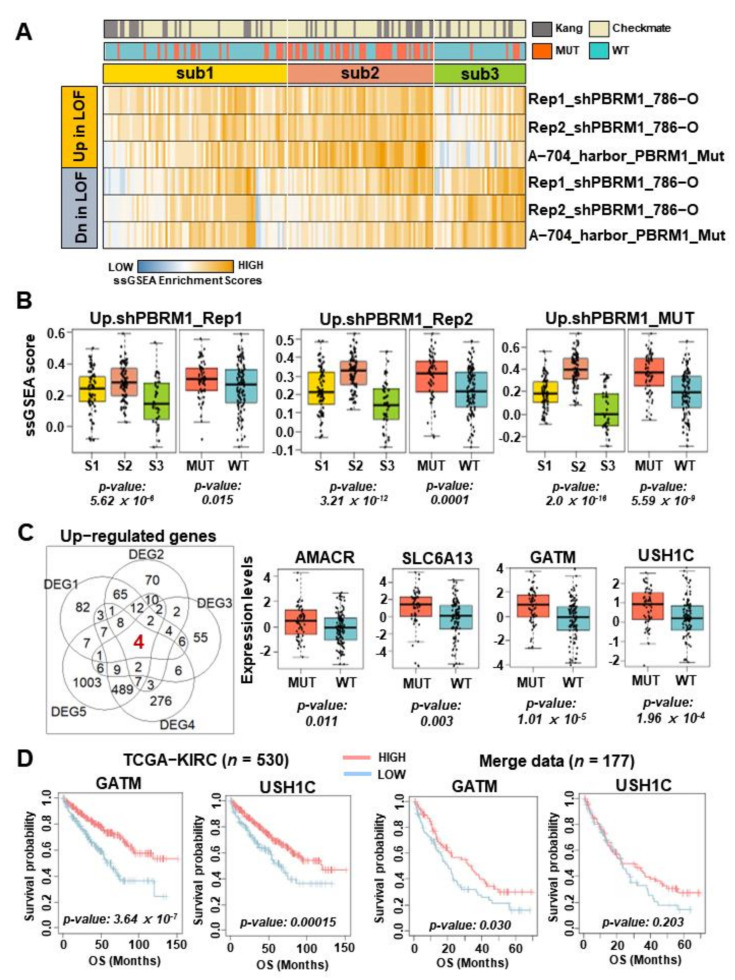
*GATM* expression is related to *PBRM1* mutation and *PBRM1* loss of function. (**A**) Heatmap of enrichment scores for the single-sample gene set enrichment analysis using upregulated and downregulated differentially expressed genes (DEGs) derived from cell line data GSE102806. (**B**) Boxplots summarizing the heatmap in (**A**) for each subtype (left) and in the group with *PBRM1* mutation vs. the group without *PBRM1* mutation (right) (S1, Sub1; S2, Sub2; S3, Sub3). (**C**) Venn diagram shows the overlapping of the five upregulated DEGs (left). Boxplots show the expression levels of *AMACR*, *SLC6A13*, *GATM*, and *USH1C* according to patients with and without *PBRM1* mutation in the merged data (right). (**D**) Kaplan–Meier plots show the overall survival of the patients stratified according the above or below the average *GATM* or *USH1C* in TCGA-KIRC (*n* = 530) and the merged data (*n* = 177). High expression (pink), low expression (sky blue).

**Figure 5 cancers-14-02354-f005:**
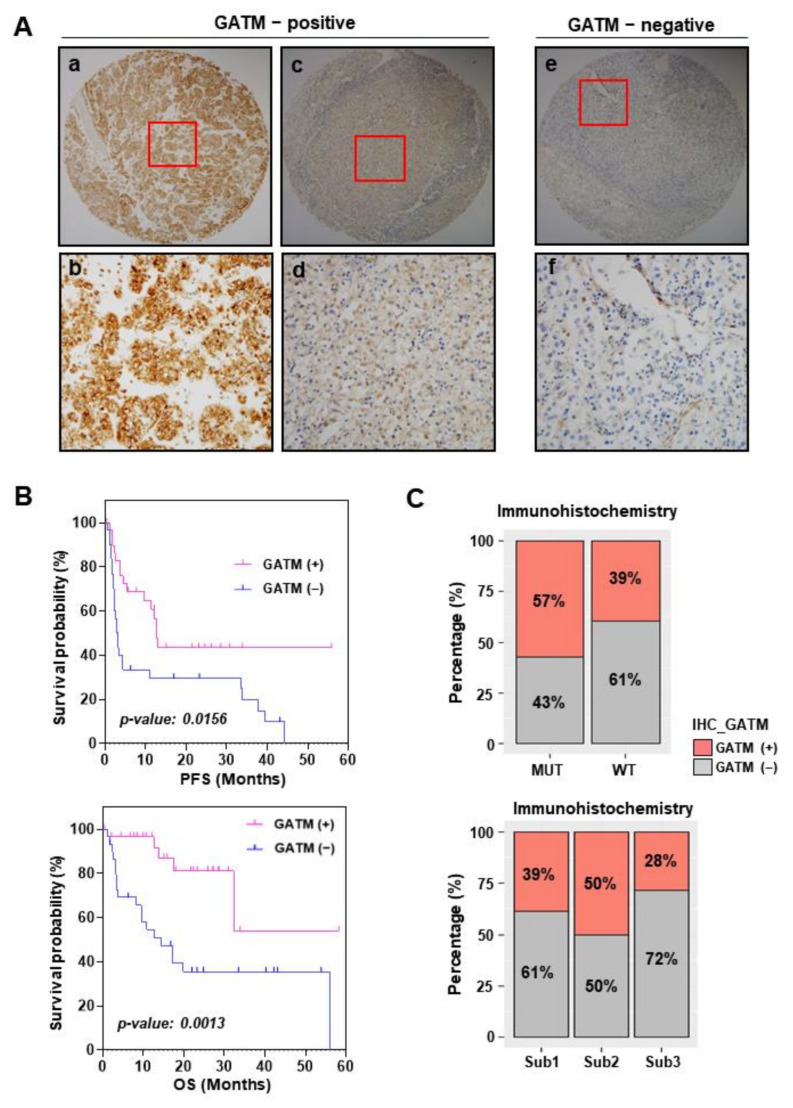
Status of GATM protein expression in immunohistochemistry are related to favorable survival. (**A**) Immunohistochemistry of GATM expression using tissue microarray of 51 patients with metastatic clear cell renal cell carcinoma treated by immune checkpoint blockades. The GATM-positive group includes “high expression” of GATM (a ×4, and b; ×20) and “low expression” of GATM (c; ×4, and d; ×20). The GATM-negative group includes “no expression” of GATM (e; ×4, and f; ×20). (**B**) Kaplan–Meier plot analysis of progression-free survival and overall survival based on the status of GATM protein expression. GATM (+) = GATM-positive group; GATM (−) = GATM-negative group. (**C**) Barplots show the percentage of both GATM-positive (+, crimson), and GATM-negative groups (−, gray), respectively, in the patients with *PBRM1* mutation vs. the patients without *PBRM1* mutation (top), and each subtype (bottom).

**Figure 6 cancers-14-02354-f006:**
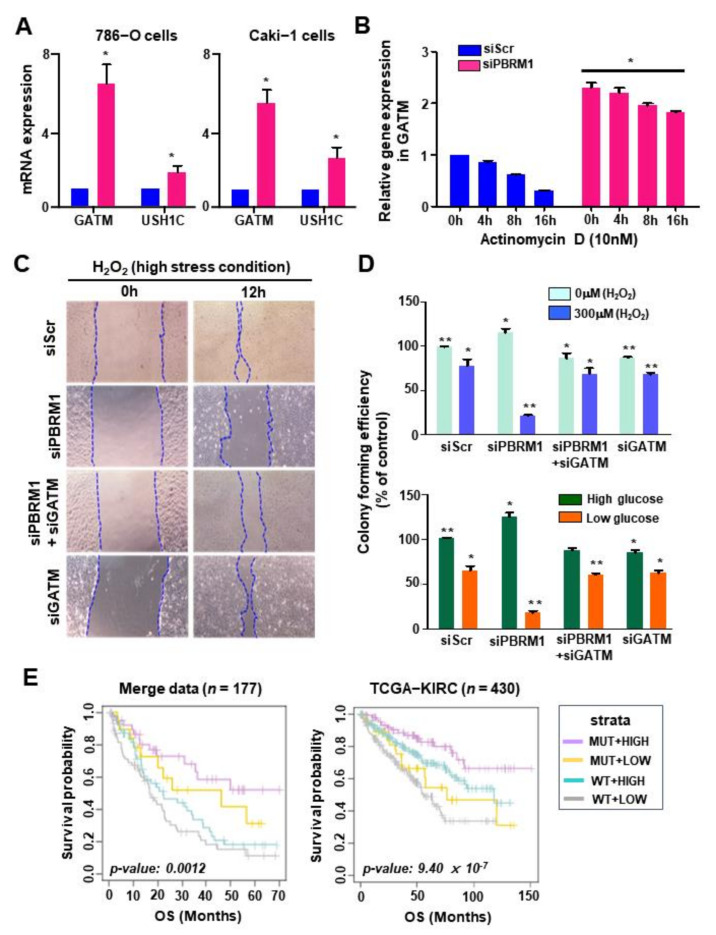
Expression of *GATM* by *PBRM1* knockdown reduces proliferation of clear cell renal cell carcinoma cell lines under stress conditions. (**A**) *GATM* or *USH1C* expression levels in PBRM1 knockdown 786-O or Caki-1 cells. The 786-O or Caki-1 cells were transfected with siScr or siPBRM1 and qRT-PCR analysis was performed. (**B**) *GATM* expression levels in siScr or siPBRM1 were transfected into 786-O cells and treated with 0.5 µM ActD for the indicated time. (**C**) Wound healing assay was performed. The 786-O cells transfected with indicated siRNAs were incubated with medium containing 300 µM H_2_O_2_ for 12 h. (**D**) Colony formation was performed in *PBRM1* knockdown 786-O cells. The 786-O cells were transfected with the indicated siRNAs and incubated with a medium containing 300 µM H_2_O_2_ or with a low-glucose medium (**E**) Kaplan–Meier plot analysis of overall survival based on *PBRM1* mutation and *GATM* expression using the merge data (*n* = 177, left). PBRM1_MUT+HIGH_GATM (*n* = 42, purple), PBRM1_MUT+LOW_GATM (*n* = 19, yellow), PBRM1_WT+HIGH_GATM (*n* = 55, blue), and PBRM1_WT+LOW_GATM (*n* = 61, gray). Kaplan–Meier plot analysis of overall survival based on *PBRM1* mutation and *GATM* expression using TCGA-KIRC (*n* = 430, right). PBRM1_MUT+HIGH_GATM (*n* = 92, purple), PBRM1_MUT+LOW_GATM (*n* = 46, yellow), PBRM1_WT+HIGH_GATM (*n* = 161, blue), and PBRM1_WT+LOW_GATM (*n* = 131, gray). The *p*-value was calculated using Student’s *t*-test: * *p* < 0.05 and ** *p* < 0.01. Data represent means ± SD from a representative experiment of at least two independent repeats (**A**,**B**,**D**).

**Figure 7 cancers-14-02354-f007:**
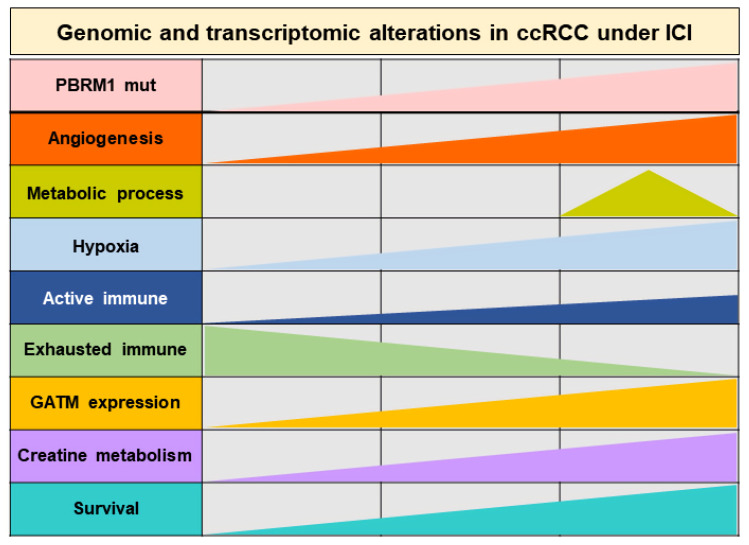
Graphical summary of the dynamics of genomic and transcriptomic aberrations in patients with metastatic clear cell renal cell carcinoma.

**Table 1 cancers-14-02354-t001:** Univariate and multivariate analyses for predicting the overall survival of patients with metastatic clear cell renal cell carcinoma treated with immune checkpoint inhibitors.

Variables		Univariate Analysis		Multivariate Analysis	
		HR (95% CI)	*p*-Value	HR (95% CI)	*p*-Value
Age	<Median	Reference	-	Reference	-
	≥Median	0.855 (0.58–1.26)	0.428	0.895 (0.599–1.339)	0.589
Sex	Female	Reference	-	Reference	-
	Male	2.047 (1.235–3.394)	0.005 **	1.576 (0.939–2.645)	0.085
IMDC	Favor	Reference	-	Reference	-
	Intermediate & Poor	2.134 (1.336–3.409)	0.0015 **	2.012 (1.252–3.233)	0.004 **
PBRM1 & GATM	MUT & HIGH	Reference	-	Reference	-
	MUT & LOW, WT & HIGH, and WT & LOW	2.532 (1.414–4.532)	0.0017 **	2.067 (1.147–3.726)	0.016 *

HR, Hazard Ratio; CI, Confidence Interval; IMDC, International Metastatic RCC Database Consortium; MUT, PBRM1 mutation; WT, PBRM1 wildtype; HIGH, GATM high expression; LOW, GATM low expression (* *p* < 0.05, ** *p* < 0.005).

## Data Availability

Whole transcriptomic sequencing data generated in this study are available from the corresponding author in this study (dr.minyong.kang@gmail.com) upon request.

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
