# Peer review of "Molecular Subtypes Based on Genomic and Transcriptomic Features Correlate with the Responsiveness to Immune Checkpoint Inhibitors in Metastatic Clear Cell Renal Cell Carcinoma"

_cancers, 2022, doi:10.3390/cancers14102354_

Round 1
Reviewer 1 Report
The study by Jee et al., is a very well-designed and organised study which is very comprehensively presented. The study contains lots of data and provide evidence supporting the conclusions. This study is based on genomic and transcriptomic data that point PBRM1 mutations, metabolic processes, immune subtypes, and GATM expression as important indicators of the response to ICI therapy in patients with metastatic ccRCC. The data further support the potential prognostic value of the PBRM1 mutation and GATM expression in patients with metastatic ccRCC treated with ICI. It provides a good reference for the conduction of future translational studies to further validate their findings and in investigate on the interacting mechanisms between PBRM1 mutation and GATM expression in the context of ICI responsiveness.
- line 235: delete "dnjs"
- Mind to use italics throughout the text when referring to gene names.
Author Response
Reviewer #1;
1) Reviewer’s comment: The study by Jee et al., is a very well-designed and organised study which is very comprehensively presented. The study contains lots of data and provide evidence supporting the conclusions. This study is based on genomic and transcriptomic data that point PBRM1 mutations, metabolic processes, immune subtypes, and GATM expression as important indicators of the response to ICI therapy in patients with metastatic ccRCC. The data further support the potential prognostic value of the PBRM1 mutation and GATM expression in patients with metastatic ccRCC treated with ICI. It provides a good reference for the conduction of future translational studies to further validate their findings and in investigate on the interacting mechanisms between PBRM1 mutation and GATM expression in the context of ICI responsiveness.
line 235: delete "dnjs"
Mind to use italics throughout the text when referring to gene names.
Author’s response: We appreciate the reviewer’s comments and have therefore revised manuscript accordingly. We deleted "dnjs" line 235 and used italics throughout the text when referring to gene name.
Reviewer 2 Report
The authors perform a comprehensive genomic analysis of patients with renal cell carcinoma undergoing immunotherapy with a special focus on PBRM1 mutations. However, some corrections and improvements have to be performed to accept the manuscript for publication.
Major:
- The methodology is correctly used and a comprehensive analysis performed. However, the clinical rational for this retrospective analysis should be emphasized if the findings should be hypothesis generating.
- The whole analysis is based on tissue that is probably sampled several years prior the respective therapy line. This fact is vaguely mentioned in the limitations section. Can the authors comment on the time frame between sampling and the therapy line and whether they see an impact on the findings? Have you data on tissue from metastatic sites? How susceptible are the observed genomic and transcriptomic features to a VEGF (first-line) therapy? This sampling issue might undermine the whole hypothesis of the manuscript and has to be considered in greater detail...
Minor:
- Line 98: CB, NCB etc is not defined in line 98 but then in line 214. Please define abbreviations when they are used for the first time throughout the manuscript.
- line 235: what is the meaning of "dnjs"
- Please carefully check English language and abbreviations.
Author Response
Reviewer #2;
2) Reviewer’s comment: The authors perform a comprehensive genomic analysis of patients with renal cell carcinoma undergoing immunotherapy with a special focus on PBRM1 mutations. However, some corrections and improvements have to be performed to accept the manuscript for publication.
Major:
The methodology is correctly used and a comprehensive analysis performed. However, the clinical rational for this retrospective analysis should be emphasized if the findings should be hypothesis generating.
The whole analysis is based on tissue that is probably sampled several years prior the respective therapy line. This fact is vaguely mentioned in the limitations section. Can the authors comment on the time frame between sampling and the therapy line and whether they see an impact on the findings? Have you data on tissue from metastatic sites? How susceptible are the observed genomic and transcriptomic features to a VEGF (first-line) therapy? This sampling issue might undermine the whole hypothesis of the manuscript and has to be considered in greater detail...
1) Can the authors comment on the time frame between sampling and the therapy line and whether they see an impact on the findings?
Author’s response: We appreciate your valuable comments to improve the quality of our study. As you pointed out, the population of this study included either first-line therapy (n = 8) or more than second-line (second-, third- or fourth-line) treatments failed to TKI (n=52). While sampling time point was at the time of diagnosis (biopsy or surgical specimen), the analysis time point was at the time of immunotherapy. Therefore, all analyzed samples were treatment-naïve status, whereas most analyzed patients were treatment-exposure status to TKIs (more than one line). It means that there was time gap between genomic/ transcriptomic status and ICB treatment response status. Actually, the selective pressure of multiple lines of TKIs treatment could increase the genomic complexity of tumors and tumor microenvironments, thus influencing responsiveness to ICB therapy. We acknowledge this critical limitation in the discussion section of the revised manuscript.
2) Have you data on tissue from metastatic sites?
Author’s response: Thank you for your comment. We extracted genomic DNA and total RNA from formalin-fixed paraffin-embedded tissues of primary kidney tumor sites (biopsy or surgical specimen from nephrectomy).
3) How susceptible are the observed genomic and transcriptomic features to a VEGF (first-line) therapy?
Author’s response: Thank you for pointing out an important issue. Our study focused on the molecular features associated with immune checkpoint blockade (ICB), not TKIs. Nevertheless, we would consider whether our molecular classification is susceptible to previous VEGF TKI. As mentioned previously, the population of this study included either first-line therapy (n = 8) or more than second-line (second-, third- or fourth-line) treatments failed to TKI. It means that patients treated by anti-PD1 monotherapy as more than second-line therapy were so heterogeneous populations regarding lines of TKIs, and therefore; we cannot obtain sufficient sample size to identify the genomic and transcriptomic features associated with responsiveness to VEGF (first-line) therapy.
Minor:
Line 98: CB, NCB etc is not defined in line 98 but then in line 214. Please define abbreviations when they are used for the first time throughout the manuscript.
line 235: what is the meaning of "dnjs"
Please carefully check English language and abbreviations.
Author’s response: We thank the reviewer’s comment. We have revised the manuscript accordingly. We added full names of CB and NCB line 98 and We also deleted "dnjs" line 235.
Reviewer 3 Report
PD-1 immunotherapy has been proved to be an effective way to treat metastatic ccRCC. However, there are still over 60% of patients could not benefit from this therapy. So, finding the reliable molecular biomarkers and prognostic signatures is very important to predict and guide the treatment strategies. In this study the authors studied the molecular subtypes Correlated with the responsiveness to immune check- 3 point Inhibitors in mccRCC based on the expression of PBRM1. They also found a novel gene GATM, which might be another biomarker in prediction of immunotherapy treatment efficacy. The paper used both clinical, biological and bioinformatic approaches to support their conclusion. Overall, the study is novel and all the data are supportive. The paper is also well written.
Author Response
Reviewer #3;
3) Reviewer’s comment: PD-1 immunotherapy has been proved to be an effective way to treat metastatic ccRCC. However, there are still over 60% of patients could not benefit from this therapy. So, finding the reliable molecular biomarkers and prognostic signatures is very important to predict and guide the treatment strategies. In this study the authors studied the molecular subtypes Correlated with the responsiveness to immune check-3 point Inhibitors in mccRCC based on the expression of PBRM1. They also found a novel gene GATM, which might be another biomarker in prediction of immunotherapy treatment efficacy. The paper used both clinical, biological and bioinformatic approaches to support their conclusion. Overall, the study is novel and all the data are supportive. The paper is also well written.
Author’s response: Thank you for the reviewer's compliment and encouragement of our study.